# Mechanistic Modelling of Biomass Growth, Glucose Consumption and Ethanol Production by *Kluyveromyces marxianus* in Batch Fermentation

**DOI:** 10.3390/e25030497

**Published:** 2023-03-14

**Authors:** Yolocuauhtli Salazar, Paul A. Valle, Emmanuel Rodríguez, Nicolás O. Soto-Cruz, Jesús B. Páez-Lerma, Francisco J. Reyes-Sánchez

**Affiliations:** 1Postgraduate Program in Engineering, Tecnológico Nacional de México/IT Durango, Blvd. Felipe Pescador 1830 Ote., Durango 34080, Mexico; 2Postgraduate Program in Engineering Sciences, BioMath Research Group, Tecnológico Nacional de México/IT Tijuana, Blvd. Alberto Limón Padilla s/n, Tijuana 22454, Mexico; 3Departamento de Ingenierías Química y Bioquímica, Tecnológico Nacional de México/IT Durango, Blvd. Felipe Pescador 1830 Ote., Durango 34080, Mexico

**Keywords:** asymptotic stability, batch fermentation, in silico experimentation, *Kluyveromyces marxianus*, nonlinear data fitting, nonlinear mechanistic model

## Abstract

This paper presents results concerning mechanistic modeling to describe the dynamics and interactions between biomass growth, glucose consumption and ethanol production in batch culture fermentation by *Kluyveromyces marxianus* (*K. marxianus*). The mathematical model was formulated based on the biological assumptions underlying each variable and is given by a set of three coupled nonlinear first-order Ordinary Differential Equations. The model has ten parameters, and their values were fitted from the experimental data of 17 *K. marxianus* strains by means of a computational algorithm design in Matlab. The latter allowed us to determine that seven of these parameters share the same value among all the strains, while three parameters concerning biomass maximum growth rate, and ethanol production due to biomass and glucose had specific values for each strain. These values are presented with their corresponding standard error and 95% confidence interval. The goodness of fit of our system was evaluated both qualitatively by in silico experimentation and quantitative by means of the coefficient of determination and the Akaike Information Criterion. Results regarding the fitting capabilities were compared with the classic model given by the logistic, Pirt, and Luedeking–Piret Equations. Further, nonlinear theories were applied to investigate local and global dynamics of the system, the Localization of Compact Invariant Sets Method was applied to determine the so-called localizing domain, i.e., lower and upper bounds for each variable; whilst Lyapunov’s stability theories allowed to establish sufficient conditions to ensure asymptotic stability in the nonnegative octant, i.e., $R_{+,0}^{3}$. Finally, the predictive ability of our mechanistic model was explored through several numerical simulations with expected results according to microbiology literature on batch fermentation.

## 1. Introduction

Alcoholic fermentation is an anaerobic process that transforms sugars like glucose or fructose into ethanol and carbon dioxide. Several yeast species are used commonly in this process, e.g., *Kloeckera, Hanseniaspora, Candida, Pichia, Kluyveromyces*, and *Saccharomyces* among others. The growth rate of these microorganisms has an ultimate effect on the sensorial characteristics of the final product, which can be positive or negative depending on the yeast used [1].

Overall, yeasts are indispensable for biotechnological processes such as wine and beer production [2]. In this research, we focus on investigating glucose consumption and ethanol production from several strains of *Kluyveromyces marxianus* (*K. marxianus*). This yeast has a great potential for alcoholic fermentation due to its intraspecific characteristics such as higher specific growth rates, the ability to grow on various substrates, and tolerance to high temperatures [3,4,5]. Further, *Kluyveromyces* sp. produces aromatic compounds such as fruity esters, carboxylic acids, ketones, furans, and other alcohols in liquid fermentation such as 2-phenyl ethanol whose sensorial characteristics can influence the quality of wine, distilled drinks, and fermented foods [6]; refer to Fonseca et al. for an extensive review on the biotechnological potentials of *K. marxianus* [4,6].

Concerning industrial applications, fermentation is commonly performed in batch culture, which brings certain advantages such as the reduction of contamination risk, in addition to the fact that a large capital investment is not necessary since high-priced production equipment is not required compared to a continuous culture process [7]. Batch process implies that yeasts are incubated in a closed container under controlled conditions with a culture medium composed of the necessary nutrients [8]. Hence, biomass cannot grow indefinitely and four phases have been identified in its dynamics, i.e., lag phase, exponential growth, stationary state, and death phase. While this process is carried out, the substrate is consumed and converted into the product, e.g., ethanol produced by sugars such as glucose [9]. Therefore, properly identifying the time interval of these phases as well as predicting the maximum product concentration that could be produced from the initial concentrations of both substrate and biomass may help to optimize production costs on the resulting product of several applications. The latter may be achieved by mechanistic modeling through predictive microbiology, which can be considered a powerful tool to investigate and summarize the overall effects of varying conditions and environmental factors within food formulation and processing [10]. Further, mathematical models could aid in gaining insights concerning microbial food safety and quality assurance of increasingly complex food products [11,12], as well as estimating shelf life and forecasting food spoilage [13,14].

Mathematical models in predictive microbiology can be classified according to different criteria, uses, and functionalities that are not mutually exclusive. Based on the type and number of variables, models are classified into primary, secondary, and tertiary; they can also be differentiated on the basis of their mathematical background as mechanistic or empirical [15], and they can be categorized into structured and unstructured conforming to the complexity of the chemical compounds of the biomass [16]. Primary models are those that represent biomass growth dynamics as a function of time, the main equations in the literature are the exponential functions of Gompertz [17] and Vazquez-Murado [18], the logarithmic function of Baranyi et al. [19] and the cubic model of Garcia et al. [20]. All models are described by parameters such as maximum growth rate $\left[\mu_{\max}\right]$, lag time $\left[L\right]$, and both initial $\left[X_{0}\right]$ and maximum biomass $\left[X_{\max}\right]$ Concentrations, while secondary models relate to the latter with environmental conditions such as temperature and pH, and other variables such as substrate and product concentrations over time, e.g., equations of Monod [21], Teisser [22], Haldane [23], and Moser [24], which aim to describe biomass growth dynamics as a function of substrate concentration and have been widely used to investigate bacterial growth [25]. Tertiary models are the result of combining primary and secondary models through the use of computer tools that allow predictions regarding the growth or death of microorganisms in food when different environmental conditions are combined [26]. Concerning the second classification mentioned, mechanistic models are formulated by means of theoretical bases and provide an interpretation of microbial growth in terms of known processes and empirical models are usually composed of polynomials of the first or second degree and pragmatically describe the data with convenient mathematical relationships, this does not usually give information on precise responses of microorganisms, because they do not take into account known processes [27]. Finally, according to the third category described, unstructured models consider biomass only as a chemical compound in a culture and its dynamics is described by simple models, while structured models also take into account changes in the internal cellular structure of biomass in terms such as the content of RNA, enzymes, reagents, metabolism and products [28]. The Gompertz, Vazquez-Murado, Baranyi and Garcia models, mentioned above, are also classified as unstructured models since biomass is considered a variable described only by its concentration. Mathematical models used by Sansonetti [29], Lei [30], and Steinmeyer [31] are classified as structured because they describe the growth of biomass considering the intracellular reactions produced by its metabolism.

Thus, it is important to highlight that in a batch fermentation process, multiple reactions occur, so the adaptability and evolution of microorganisms in short periods and changes in environmental conditions usually characterize this type of process, consequently, the modeling of these systems is complex due to time-varying characteristics of biological systems, resulting in nonlinear systems dynamics [28]. Hence, a mathematical model formulated from a system of nonlinear differential equations will allow the application of nonlinear systems control methods to optimize the process so that the characteristics of the final product can be predicted when the environmental conditions of the culture are controlled and the initial conditions of biomass, substrate, and product values are known. It is worth mentioning that most of the models found in the literature focus on the yeast *Saccharomyces cerevisiae* since it is one of the most used in the industry; however, biotechnological opportunities have been found in non-Saccharomyces yeasts because they have metabolic characteristics that lead to the production of compounds of interest. Therefore, it is important to model the growth of *K. marxianus* because of the great potential in the production of esters compounds of industrial importance [32]. Thus, in this paper, we applied mechanistic and computational modeling to formulate a system of three coupled nonlinear first-order Ordinary Differential Equations (ODEs) that describe dynamics between biomass, glucose (substrate), and ethanol (product) concentrations over time. Mechanistic modeling allowed us to provide both qualitative and quantitative descriptions concerning the relationships of biomass growth, glucose consumption, and ethanol production from 17 strains of *K. marxianus*, while computational modeling was used to fit experimental data from these three variables and establish numerical values for each parameter of the mathematical model. Further, nonlinear theories such as the Localization of Compact Invariant Sets (LCIS) method and Lyapunov’s Stability Theory were applied to provide a complete analysis of the local and global dynamics of our proposed biological system [33].

## 2. Materials and Methods

This section provides all the information concerning the experimental data of biomass growth, substrate consumption and ethanol production, i.e., karyotypes of the *K. marxianus* strains with identifiable chromosomal differences among them, environmental conditions, chemical characteristics of the medium, lab equipment used for measurements, and periods for each measurement, then the mathematical model is formulated and each equation as well as values and units of parameters are described. This section concludes by describing the procedure to approximate the experimental data and to fit the numerical values of each parameter by designing an algorithm in Matlab.

### 2.1. Experimental Data: Culture Medium and Analytical Techniques

Experimental data was obtained from alcoholic fermentation in batch culture by *K. marxianus*, 17 strains with different genetic profiles were incubated in 20 g/L of yeast extract peptone dextrose agar at $30{\;}^{\circ }$C in order to study their kinetic growth, glucose consumption and ethanol production. Codification and origin of studied karyotypes of *K. marxianus* are identified by Páez et al. in [34], where 15 strains were obtained from different places of México, and they were isolated from agave fermentation for mezcal production, in addition to 2 reference strains that were isolated from pozol (CBS6556) in México, and from yoghurt (CBS397) in Netherlands.

Characteristics of the chemically defined medium are given as follows: glucose 20 g/L, KH${}_{2}$PO${}_{4}$ 3 g/L, (NH${}_{4}$)${}_{2}$SO${}_{4}$ 3 g/L, Na${}_{2}$HPO${}_{4}$ 1.49 g/L, glutamic acid 1 g/L, MgCl${}_{2}$ heptahydrate 0.41 g/L, ZnCl${}_{2}$ 0.0192 g/L, CuCl${}_{2}$ 0.0006 g/L, MnCl${}_{2}$ 0.044 g/L, CoCl${}_{2}$ 0.0005 g/L, CaCl 0.0117 g/L, FeCl${}_{2}$ 0.011 g/L, (NH${}_{4}$)${}_{6}$Mo${}_{7}$O${}_{24}$ 0.004 g/L, H${}_{3}$BO${}_{4}$ 0.0030 g/L, aminobenzoic acid 0.0010 g/L, myo-inositol 0.1250 g/L, nicotinic acid 0.0050 g/L, pantothenic acid 0.005 g/L, pyridoxine 0.0050 g/L, thiamin HCl 0.005 g/L, biotin 0.000024 g/L [35]. This medium was used to culture the strains for biomass development with agitation, for the conservation of the strains, plates with the same medium with 20% agar were used and stored at $4{\;}^{\circ }$C.

Biomass concentration was measured with a spectrophotometer UV-VIS DR 6000 (HACH, Loveland, CO, USA) by optical density at 600 nm, values in g/L were obtained relating optical density with a calibration curve of the dry weight of *K. marxianus*. For glucose consumption and ethanol production by High-Performance Liquid Chromatography (HPLC series 1200, Agilent Technology, Palo Alto, CA, USA), a BIORAD HP-$87H^{+}\left(8\%\right)$ ion exchange column was used, in an AGILENT® 1200 series equipment, with H${}_{2}$SO${}_{4}$ 0.005 N as mobile phase, at a flow of 0.5 mL/min, the column temperature was $60{\;}^{\circ }$C, and the Refractive Index detector temperature was $60{\;}^{\circ }$C. The injection volume of 5 μL, calibration curves were made with glucose and ethanol Sigma Aldrich  at 99% purity or higher, and a determination coefficient higher than 0.99 for each compound [36,37].

Fermentation was made in duplicate for every strain and samples were taken each hour for 13 consecutive hours. Two samples were taken every hour for each variable in the time interval of the process where *t* goes from 0 to 13, then the average value of the two measurements was computed. Therefore, each variable, i.e., biomass $\left[x(t)\right]$, glucose $\left[y(t)\right]$, and ethanol $\left[z(t)\right]$, has 14 observations with a total of 42 experimental data points $\left(n\right)$ for each *K. marxianus* strain.

### 2.2. The KM Mechanistic Model

The KM mechanistic model is proposed to describe the dynamics of alcoholic fermentation. This is a biochemical process carried out by yeasts (also known as biomass), to transform sugars such as glucose into ethyl alcohol, otherwise known as ethanol (main product) and other byproducts. In this case, the alcoholic fermentation is taken in a batch fermentation process with established laboratory conditions of temperature and known initial glucose concentrations (substrate). Our mathematical model describes the relationships between biomass concentration $\left[x\left(t\right)\right]$, glucose consumption $\left[y\left(t\right)\right]$, and ethanol production $\left[z\left(t\right)\right]$ over time. The set of three first-order ODEs is presented below
(1)$$\begin{array}{ccc} \dot{x} & = & \frac{\rho_{1}xy}{\rho_{2}+y}-\rho_{3}xz-\rho_{4}x, \end{array}$$
(2)$$\begin{array}{ccc} \dot{y} & = & -\rho_{5}xy-\rho_{6}yz-\rho_{7}y, \end{array}$$
(3)$$\begin{array}{ccc} \dot{z} & = & \rho_{8}xz+\rho_{9}yz-\rho_{10}z, \end{array}$$
where each state variable $x\left(t\right)$, $y\left(t\right)$ and $z\left(t\right)$ are measured in g/L, and the time unit is given in hours. Now, by considering results from Leenheer and Aeyels (see Section II.A in [38]), all solutions with nonnegative initial conditions $\left[x\left(0\right),y\left(0\right),z\left(0\right)\geq 0,\right]$ will be located in the nonnegative octant as indicated below
$$R_{+,0}^{3}=\left\{x\left(t\right),y\left(t\right),z\left(t\right)\geq 0\right\},$$
i.e., each positive half trajectory of the system will be positively forward invariant in $R_{+,0}^{3}$. The latter also considers the biological sense of each variable as there is no meaning for negative values of biomass, glucose or ethanol in the scope of the KM system (Equation 1)–(Equation 3). It is important to mention that variables cannot grow exponentially indefinitely, and they must have biologically feasible limits which will be discussed in the next section. Values and units of each parameter of the KM system (Equation 1)–(Equation 3) are given in Table 1.

Now, let us describe our mechanistic model based on the experimental data described in the previous section and the following biological assumptions. Biomass growth dynamics is described by Equation (Equation 1), where the first term uses the classical Monod form for the growth of microorganisms [21], where $\rho_{1}$ is the biomass maximum growth rate (also found in the literature as $\mu_{\max}$), and $\rho_{2}$ is the affinity or half-velocity constant for glucose consumption. The second term describes biomass death due to ethanol accumulation toxicity by the law of mass action (see Section 2.3 in [39]) with a rate $\rho_{3}$. This term is negative because ethanol accumulation increases the membrane fluidity and negatively affects the membrane protein’s function, which can lead to cell growth inhibition or even death [40,41]. The third term represents the natural yeast death rate $\left[\rho_{4}\right]$ mainly due to environmental resources depletion [42]. Glucose dynamics is formulated in Equation (Equation 2) as a decrescent function where the law of mass action gives the first two terms. The first one represents glucose consumption to support biomass growth. In contrast, the second term accounts for the glucose consumption used for ethanol production, with rates $\rho_{5}$ and $\rho_{6}$, respectively. The third term represents the spontaneous decomposition rate of glucose $\left[\rho_{7}\right]$ [43]. The latter stems from the fact that the culture medium is placed in a sealed container in batch fermentation, and no other nutrients (primarily glucose) are supplied into the system. Ethanol dynamics is described in Equation (Equation 3). The first term represents ethanol production associated with biomass growth. It is due to ethanol being recognized as a primary metabolite, a product obtained from reactions required or cellular growth [44,45]. The second term represents the glucose conversion to ethanol not directly linked to cellular growth, attributed to the need for Gibbs’s free energy for cellular maintenance [44,46]. In both cases, terms are formulated by the law of mass action with respective rates $\rho_{8}$ and $\rho_{9}$. Finally, the third term represents ethanol degradation with a rate $\rho_{10}$. The flow diagram shown in Figure 1 was constructed to illustrate the dynamics of the system.

It should be noted that fixed parameters values were estimated for the 17 *K. marxianus* strains, particularly for $\rho_{2}$, $\rho_{3}$, $\rho_{4}$, $\rho_{5}$, $\rho_{6}$, $\rho_{7}$ and $\rho_{10}$. Further, concerning the death rate of biomass $\left[\rho_{4}\right]$, spontaneous decomposition rate of glucose $\left[\rho_{7}\right]$, and degradation rate of ethanol $\left[\rho_{10}\right]$, one can see that they are in a different order of magnitude and the following constraint is formulated for these three parameters:(4)$$\rho_{10}>\rho_{4}>\rho_{7}.$$

Now, concerning the equilibrium points of the KM system, Equations (Equation 1)–(Equation 3) have a unique biologically meaningful equilibrium point in the domain $R_{+,0}^{3}$ given by
(5)$$\left(x_{0}^{*},y_{0}^{*},z_{0}^{*}\right)=\left(0,0,0\right).$$

Another set of five equilibria with at least one negative value is shown in Appendix B; therefore, these equilibrium points are discarded from the biologically meaningful dynamics of the system. Further, from the biological characteristics of each variable the following can be stated with respect to each solution as time increases
$$\lim_{t\to \infty }x\left(t\right)=\lim_{t\to \infty }y\left(t\right)=\lim_{t\to \infty }z\left(t\right)=0,$$
due to the eventual death of microorganisms, glucose consumption and ethanol degradation [47], asymptotic stability of trajectories is discussed in the next section.

### 2.3. Parameter Value Estimation

First, let us compute the glucose decomposition rate $\left[\rho_{7}\right]$ by assuming a first-order kinetics [48] for glucose dynamics, and considering a half-life $\left[t_{1/2}\right]$ of 96 years [43]. Then, $\rho_{7}$ can be computed from the next equation
$$y\left(t\right)=y_{0}e^{-\rho_{7}t},$$
as follows
$$\frac{y_{0}}{2}=y_{0}e^{-\rho_{7}t_{1/2}},$$
where $y_{0}$ is the glucose initial concentration, i.e., $y\left(0\right)$; hence
$$\rho_{7}=\frac{\ln2}{t_{1/2}}=824.233\times 10^{-9}\;h^{-1}.$$

Now, in order to determine the numerical values of parameters $\rho_{i}$, i=1,2,3,4,5,6,8,9, 10; the computational model of Equations (Equation 1)–(Equation 3) was formulated as follows
(6)$$\begin{array}{ccc} x_{i+1} & = & x_{i}+\left(\frac{\rho_{1}x_{i}y_{i}}{\rho_{2}+y_{i}}-\rho_{3}x_{i}z_{i}-\rho_{4}x_{i}\right)\Delta_{t}, \end{array}$$
(7)$$\begin{array}{ccc} y_{i+1} & = & y_{i}+\left(-\rho_{5}x_{i}y_{i}-\rho_{6}y_{i}z_{i}-\rho_{7}y_{i}\right)\Delta_{t}, \end{array}$$
(8)$$\begin{array}{ccc} z_{i+1} & = & z_{i}+\left(\rho_{8}x_{i}z_{i}+\rho_{9}y_{i}z_{i}-\rho_{10}z_{i}\right)\Delta_{t}, \end{array}$$
by applying Euler’s method (see Section 1.7 in [49]) where $\Delta_{t}$ was set to $1\times 10^{-5}$. Then, an algorithm was formulated in Matlab 2022b with the *lsqcurvefit* function from the optimization toolbox as its core [50] (initial points were set as $1\times 10^{-1}$ for each parameter $\left[\rho_{1},\rho_{8},\rho_{9}\right]$, i.e., $x_{0}=\left[1\times 10^{-1};1\times 10^{-1};1\times 10^{-1}\right]$, and optimotions of the function were set as follows: Max Function Evaluations $=1\times 10^{3}$, Max Iterations $=1\times 10^{3}$, and Function Tolerance $=1\times 10^{-9}$). This allowed us to establish a fixed value for parameters $\rho_{j}$, j=2,3,4,5,6,10; by averaging the corresponding values for each of the 17 strains, these results are shown in Table 1. However, this procedure was not applicable for parameters $\rho_{1}$, $\rho_{8}$ and $\rho_{9}$, as it was expected that each strain of *K. marxianus* will have its own biomass growth rate $\left[\rho_{1}\right]$, and its corresponding ethanol production rates $\left(\rho_{8}\;\mathrm{and}\;\rho_{9}\right)$, this is directly linked to the chromosomal differences among the strains affecting their growth kinetics. Hence, the main algorithm was redesigned to fit these three parameters and consider the others as fixed constants. Overall results are shown in Table 2 with their corresponding standard error (SE), and 95% confidence interval (CI). These two statistics allow us to establish that estimates for parameters $\rho_{1}$, $\rho_{8}$, and $\rho_{9}$ in the 17 strains are statistically significant. The latter follows from the fact that each $SE\left(\rho_{k}\right)<\rho_{k}/2$, k=1,8,9; i.e., the value of the SE is less than half of the value fitted for each parameter, thus, the null hypothesis $\left[\rho_{k}=0\right]$ can be rejected (see Section 5.2.8 from Koutsoyiannis [51]). Furthermore, both the lower and upper limit of the 95% CI of all fitted values are positive, hence, as there is no change in the sign of the bounds, this implies that the value of the null hypothesis is excluded, and one can conclude that all P-values are less than 0.05 (see Chapter 17 from Motulsky [52]).

Finally, it should be noted that the in silico experimentation performed in this research was done on a high-end desktop computer with a Ryzen 9 5950X CPU, 128 GB of RAM DDR4 CL18, a 12 GB GPU NVIDIA GeForce RTX 3080, and 1 TB Samsung 980 Pro Gen 4 NVMe M.2. The complete algorithm that was designed to fit the numerical values of parameters $\left[\rho_{1},\rho_{8},\rho_{9}\right]$, and to determine results concerning the statistics and goodness of fit can be found in the Appendix A.

## 3. Results

In this section, the in silico experimentation is performed by means of several numerical simulations, and results relating to the nonlinear analysis of the system are derived, i.e., bounds for the localizing domain, asymptotic stability, and existence and uniqueness for all solutions of our model in the nonnegative octant $R_{+,0}^{3}$.

### 3.1. In Silico Experimentation and Goodness of Fit

First, qualitative results are illustrated by means of numerical simulations. For the sake of simplicity, the strains were clustered in groups of four from strain 1 to the 16 (see Figure 2, Figure 3, Figure 4 and Figure 5, respectively), and results concerning only for the strain 17 are shown in Figure 6. In all panels, the × green marker represents the average value for the two experimental data measurements for each variable, i.e., biomass $\left[x\left(t\right)\right]$, glucose $\left[y\left(t\right)\right]$, and ethanol $\left[z\left(t\right)\right]$, whilst the blue continuous line represents the approximated value given by the KM system (Equation 1)–(Equation 3) when is solved by means of Equations (Equation 6)–(Equation 8) with $\Delta_{t}=1\times 10^{-5}$. The time units are given in hours and the concentration for each variable is measured in g/L as indicated in each axis. Values for all ten parameters corresponding to each strain are shown in Table 1 and Table 2.

Now, let us provide a quantitative measure of the fitting capabilities of the KM mechanistic model (Equation 1)–(Equation 3), thus, the coefficient of determination $\left[R^{2}\right]$ is calculated for each variable with results shown in Table 3.

Further, the Akaike Information Criterion (AIC) [53,54,55] was computed by considering a small sample relative to the number of parameters $\left(n/K<40\right)$ with a bias correction as indicated below
$$AIC=n\ln\left(\frac{\sum_{i=1}^{n}{\left[f_{e}\left(i\right)-f_{a}\left(i\right)\right]}^{2}}{n}\right)+2K+\frac{2K(K+1)}{n-K-1},$$
where *n* is the total number of experimental data points; $f_{e}$ the experimental data and $f_{a}$ the approximated value for the residual sum of squares (RSS); and *K* the number of parameters of the system; therefore, $n/K=\left(3\times 14\right)/10=4.2$. Results, including RSS, AIC and $R^{2}$, for the complete trajectory of the system, i.e., $\phi \left(x,y,z\right)$ for the total of 42 experimental points (14 for each variable) are summarized in Table 4.

The AIC yields a value that relates the amount of information that our model loses when approximating the experimental data. Hence, one can compare the capabilities of the model to estimate the concentrations over time of biomass $\left[x\left(t\right)\right]$, glucose $\left[y\left(t\right)\right]$, and ethanol $\left[z\left(t\right)\right]$ among the 17 *K. marxianus* strains while providing a statistical measured for the quality of the KM system (Equation 1)–(Equation 3).

### 3.2. Nonlinear Analysis: Localizing Domain, Asymptotic Stability, Existence and Uniqueness

The localizing domain can be determined by computing the upper bounds for all variables of the KM mechanistic model (Equation 1)–(Equation 3), the lower bounds are given by the boundary of the domain $R_{+,0}^{3}$, i.e., $\left\{x_{\inf}=0,y_{\inf}=0,z_{\inf}=0\right\}$. The latter is achieved by means of integration and the LCIS method [56]. Within the localizing domain, one may find all biologically meaningful dynamics of the system, i.e., compact invariant sets such as equilibrium points, periodic orbits, limit cycles and chaotic attractors (see Section 3 in [57]), among others.

First, in order to find the upper bound for the glucose concentration $\left[y\left(t\right)\right]$, Equation (Equation 2) is integrated as follows
$$\int \frac{dy}{y}=-\int_{0}^{t}f\left(x,z\right)dt,$$
where
$$f\left(x,z\right)=\rho_{5}x+\rho_{6}z+\rho_{7}>0,$$
by considering $\rho_{7}>0$ and $x\left(t\right),z\left(t\right)\geq 0$ from the domain $R_{+,0}^{3}$. Then,
$$y\left(t\right)=y\left(0\right)\exp\left[-\int_{0}^{t}f\left(x,z\right)dt\right],$$
with $y\left(0\right)\in R_{+,0}$. Therefore, all solutions with nonnegative initial conditions will be bounded as indicated below
$$K_{y}=\left\{0\leq y\left(t\right)\leq y_{\sup}=y\left(0\right)\right\},$$
hence, any upper bound for $x\left(t\right)$ and $z\left(t\right)$ depending on $K_{y}$ will be directly related to the glucose initial concentration $\left[y\left(0\right)\right]$, which is expected as biomass and ethanol production over time is directly related to glucose dynamics.

Now, let us provide the mathematical background that allows us to compute a localizing domain where all compact invariant sets of a nonlinear dynamical system are located. The General Theorem concerning the LCIS method was formalized by Krishchenko and Starkov (see Section 2 in [58]) and it states the following: *Each compact invariant set*
Γ
*of*
$\dot{x}=f\left(x\right)$
*is contained in the localizing domain:*$$K\left(h\right)=\left\{h_{\inf}\leq h\left(x\right)\leq h_{\sup}\right\}.$$
From the latter $f\left(x\right)$ is a $C^{\infty }–$differentiable vector function where $x\in R^{n}$ is the state vector. $h\left(x\right):R^{n}\to R$ is a $C^{\infty }–$differentiable function called localizing function, ${h|}_{S}$ denotes the restriction of $h\left(x\right)$ on a set $S\subset R^{n}$ with $S\left(h\right)=\left\{x\in R^{n}|L_{f}h\left(x\right)=0\right\}$, and $L_{f}h\left(x\right)=\left(\partial h/\partial x\right)f\left(x\right)$ is the Lie derivative of $f\left(x\right)$. Hence, one can define $h_{\inf}=\inf\left\{h\left(x\right)|x\in S\left(h\right)\right\}$ and $h_{\sup}=\sup\left\{h\left(x\right)|x\in S\left(h\right)\right\}$. Furthermore, if all compact invariant sets are contained in the set $K\left(h_{i}\right)$ and in the set $K\left(h_{j}\right)$ then they are contained in $K\left(h_{i}\right)$$\cap K\left(h_{j}\right)$ as well. The nonexistence of compact invariant sets can be considered for a given set $\Lambda \subset R^{n}$ if Λ∩K(h)=∅, then the system $\dot{x}=f\left(x\right)$ has no compact invariant sets located in Λ.

Following the LCIS method, one can explore the next localizing function
$$h_{1}=x+\alpha y;\;\alpha >0,$$
then, the Lie derivative may be written as follows
$$L_{f}h_{1}=-\rho_{4}x-\alpha \rho_{7}y-\frac{\alpha \rho_{5}y+\alpha \rho_{2}\rho_{5}-\rho_{1}}{\rho_{2}+y}xy-\rho_{3}xz-\alpha \rho_{6}yz,$$
and the set $S\left(h_{1}\right)=\left\{L_{f}h_{1}=0\right\}$ is given by
$$S\left(h_{1}\right)=\left\{\rho_{4}x=-\alpha \rho_{7}y-\frac{\alpha \rho_{5}y+\alpha \rho_{2}\rho_{5}-\rho_{1}}{\rho_{2}+y}xy-\rho_{3}xz-\alpha \rho_{6}yz\right\},$$
where $x=h_{1}-\alpha y$, therefore set $S\left(h_{1}\right)$ is rewritten as indicated below
$$S\left(h_{1}\right)=\left\{h_{1}=\frac{\alpha \left(\rho_{4}-\rho_{7}\right)}{\rho_{4}}y-\frac{\alpha \rho_{5}y+\alpha \rho_{2}\rho_{5}-\rho_{1}}{\rho_{4}\left(\rho_{2}+y\right)}xy-\frac{\rho_{3}}{\rho_{4}}xz-\frac{\alpha \rho_{6}}{\rho_{4}}yz\right\},$$
and the next two conditions are formulated
(9)$$\begin{array}{ccc} \rho_{4}-\rho_{7} & > & 0, \end{array}$$
(10)$$\begin{array}{ccc} \alpha & > & \frac{\rho_{1}}{\rho_{2}\rho_{5}}, \end{array}$$
where (Equation 9) is directly fulfilled by (Equation 4). Now, let us apply the Iterative Theorem in order to find an upper bound for the localizing function
$$S\left(h_{1}\right)\cap K_{y}\subset \left\{h_{1}\leq \frac{\alpha \left(\rho_{4}-\rho_{7}\right)}{\rho_{4}}y_{\sup}\right\},$$
then
$$K\left(h_{1}\right)=\left\{x\left(t\right)+\alpha y\left(t\right)\leq \frac{\alpha \left(\rho_{4}-\rho_{7}\right)}{\rho_{4}}y_{\sup}\right\},$$
from the latter, the upper bound for the biomass concentration $\left[x\left(t\right)\right]$ may be written in terms of the parameters and the initial glucose concentration $\left[y\left(0\right)\right]$ as follows
$$K_{x}=\left\{0\leq x\left(t\right)\leq x_{\sup}=\frac{\alpha \left(\rho_{4}-\rho_{7}\right)}{\rho_{4}}y_{\sup}\right\}.$$

Now, an upper bound for the ethanol concentration $\left[z\left(t\right)\right]$ can be determined by the following localizing function
$$h_{2}=\beta_{1}x+\beta_{2}y+z;\;\beta_{1},\beta_{2}>0,$$
whose Lie derivative is computed as indicated below
$$L_{f}h_{2}=-\beta_{1}\rho_{4}x-\beta_{2}\rho_{7}y-\rho_{10}z-\frac{\beta_{2}\rho_{5}y+\beta_{2}\rho_{2}\rho_{5}-\beta_{1}\rho_{1}}{\rho_{2}+y}xy-\left(\beta_{1}\rho_{3}-\rho_{8}\right)xz-\left(\beta_{2}\rho_{6}-\rho_{9}\right)yz,$$
and at this step, the following conditions are formulated
(11)$$\begin{array}{ccc} \beta_{1} & > & \frac{\rho_{8}}{\rho_{3}}, \end{array}$$
(12)$$\begin{array}{ccc} \beta_{2} & > & \max\left\{\frac{\rho_{9}}{\rho_{6}},\frac{\beta_{1}\rho_{1}}{\rho_{2}\rho_{5}}\right\}, \end{array}$$
then, set $S\left(h_{2}\right)=\left\{L_{f}h_{2}=0\right\}$, can be written as follows
$$S\left(h_{2}\right)=\left\{\rho_{10}z=-\beta_{1}\rho_{4}x-\beta_{2}\rho_{7}y-\frac{\beta_{2}\rho_{5}y+\beta_{2}\rho_{2}\rho_{5}-\beta_{1}\rho_{1}}{\rho_{2}+y}xy-\left(\beta_{1}\rho_{3}-\rho_{8}\right)xz-\left(\beta_{2}\rho_{6}-\rho_{9}\right)yz\right\},$$
hence, as $z=h_{2}-\beta_{1}x-\beta_{2}y$, then set $S\left(h_{2}\right)$ is rewritten as indicated below
$$S\left(h_{2}\right)=\left\{h_{2}=\frac{\beta_{1}\left(\rho_{10}-\rho_{4}\right)}{\rho_{10}}x+\frac{\beta_{2}\left(\rho_{10}-\rho_{7}\right)}{\rho_{10}}y-\frac{\beta_{2}\rho_{5}y+\beta_{2}\rho_{2}\rho_{5}-\beta_{1}\rho_{1}}{\rho_{2}+y}xy-\left(\beta_{1}\rho_{3}-\rho_{8}\right)xz-\left(\beta_{2}\rho_{6}-\rho_{9}\right)yz\right\},$$
where the next condition is formulated
(13)$$\rho_{10}>\max\left\{\rho_{4},\rho_{7}\right\},$$
and it holds by (Equation 4). Then, the Iterative Theorem is applied to get the following result
$$S\left(h_{2}\right)\cap K_{x}\cap K_{y}\subset \left\{h_{2}\leq \frac{\beta_{1}\left(\rho_{10}-\rho_{4}\right)}{\rho_{10}}x_{\sup}+\frac{\beta_{2}\left(\rho_{10}-\rho_{7}\right)}{\rho_{10}}y_{\sup}\right\},$$
then, the upper bound for the localizing function $h_{2}$ is derived as follows
$$K\left(h_{2}\right)=\left\{\beta_{1}x\left(t\right)+\beta_{2}y\left(t\right)+z\left(t\right)\leq \frac{\beta_{1}\left(\rho_{10}-\rho_{4}\right)}{\rho_{10}}x_{\sup}+\frac{\beta_{2}\left(\rho_{10}-\rho_{7}\right)}{\rho_{10}}y_{\sup}\right\},$$
now, from the latter one can get the upper bound for ethanol concentration $\left[z\left(t\right)\right]$ over time in terms of the parameters, the initial glucose concentration $\left[y\left(0\right)\right]$ and the upper bound of biomass $\left[x_{\sup}\right]$ as given below
$$K_{z}=\left\{0\leq z\left(t\right)\leq z_{\sup}=\frac{\beta_{1}\left(\rho_{10}-\rho_{4}\right)}{\rho_{10}}x_{\sup}+\frac{\beta_{2}\left(\rho_{10}-\rho_{7}\right)}{\rho_{10}}y_{\sup}\right\}.$$

Results shown above allow us to conclude the following regarding the boundedness of the KM system (Equation 1)–(Equation 3) solutions:

**Theorem 1.** ***Localizing domain***. *If conditions (Equation 9)–(Equation 13) are fulfilled, then all compact invariant sets of the KM mechanistic model (Equation 1)–(Equation 3) are located either at the boundaries or within the following domain*$$K_{\Gamma }=K_{x}\cap K_{y}\cap K_{z},$$*where $K_{\Gamma }\subset R_{+,0}^{3}$, and the ultimate bounds for biomass $\left[x\left(t\right)\right]$, glucose $\left[y\left(t\right)\right]$, and ethanol $\left[z\left(t\right)\right]$ concentrations over time are given below*$$\begin{array}{ccc} K_{x} & = & \left\{0\leq x\left(t\right)\leq x_{\sup}=\frac{\alpha \left(\rho_{4}-\rho_{7}\right)}{\rho_{4}}y_{\sup}\right\},\\ K_{y} & = & \left\{0\leq y\left(t\right)\leq y_{\sup}=y\left(0\right)\right\},\\ K_{z} & = & \left\{0\leq z\left(t\right)\leq z_{\sup}=\frac{\beta_{1}\left(\rho_{10}-\rho_{4}\right)}{\rho_{10}}x_{\sup}+\frac{\beta_{2}\left(\rho_{10}-\rho_{7}\right)}{\rho_{10}}y_{\sup}\right\}. \end{array}$$

Now, let us briefly provide the mathematical background concerning the stability theory in the sense of Lyapunov, particularly the direct method where it is necessary to formulate a Lyapunov candidate function, which is usually denoted as $V\left(x\right):R^{n}\to R$, a continuously differentiable function whose temporal derivative is given by $\dot{V}\left(x\right)=\left(\partial V/\partial x\right)f\left(x\right)$. This function must be positive definite, i.e., $V\left(0\right)=0$ and $V\left(x\right)>0$ for x≠0, whilst a negative definite function is also $V\left(0\right)=0$ but $V\left(x\right)<0$ for x≠0. Further, function $V\left(x\right)$ is said to be radially unbounded if $V\left(x\right)\to \infty$ as $∥x∥\to \infty$. The latter allows the formulation of the Global Asymptotic Stability Theorem (see Chapter 4 in [59] and Chapter 2 in [60]) which states the following: *The equilibrium point*
$x^{*}$
*is globally asymptotically stable if there exists a function* $V\left(x\right)$
*positive definite, radially unbounded and decrescent such that its temporal derivative*
$\dot{V}\left(x\right)$
*is negative definite.* A function $V\left(x\right)$ satisfying properties of this theorem is called Lyapunov function.

Following the latter, let us explore the next Lyapunov candidate function $$V\left(x,y,z\right)=\gamma_{1}x+\gamma_{2}y+z,$$ with $$\gamma_{1},\gamma_{2}>0,$$ then, the time derivative is computed as shown below $$\dot{V}\left(x,y,z\right)=\gamma_{1}\left(\frac{\rho_{1}xy}{\rho_{2}+y}-\rho_{3}xz-\rho_{4}x\right)-\gamma_{2}\left(\rho_{5}xy+\rho_{6}yz+\rho_{7}y\right)+\rho_{8}xz+\rho_{9}yz-\rho_{10}z,$$
which can be rewritten as follows
$$\dot{V}\left(x,y,z\right)=-\gamma_{1}\rho_{4}x-\gamma_{2}\rho_{7}y-\rho_{10}z-\left(\gamma_{1}\rho_{3}-\rho_{8}\right)xz-\left(\gamma_{2}\rho_{6}-\rho_{9}\right)yz-\frac{y\gamma_{2}\rho_{5}+\gamma_{2}\rho_{2}\rho_{5}-\gamma_{1}\rho_{1}}{\rho_{2}+y}xy,$$
where it is evident that $\dot{V}\left(0,0,0\right)=0$, therefore the following constraints on coefficients $\gamma_{1}$ and $\gamma_{2}$ are formulated to ensure $\dot{V}\left(x,y,z\right)<0$
(14)$$\begin{array}{ccc} \gamma_{1} & > & \frac{\rho_{8}}{\rho_{3}}, \end{array}$$
(15)$$\begin{array}{ccc} \gamma_{2} & > & \max\left\{\frac{\rho_{9}}{\rho_{6}},\frac{\gamma_{1}\rho_{1}}{\rho_{2}\rho_{5}}\right\}, \end{array}$$
thus, as parameters $\rho_{i},i=1,2,3,5,6,8,9;$ in both conditions are different for each term, then it is possible to assume that there exists a set of solutions that satisfies (Equation 14) and (Equation 15). Hence, the following result can be concluded:

**Theorem 2.** ***Asymptotic stability****. If conditions (Equation 14) and (Equation 15) are fulfilled, then the KM mechanistic model (Equation 1)–(Equation 3) is asymptotically stable and all trajectories will go to the equilibrium point* 
$\left(x_{0}^{*},y_{0}^{*},z_{0}^{*}\right)=\left(0,0,0\right).$

The latter implies that any given trajectory $\left[\phi \left(x\left(t\right),y\left(t\right),z\left(t\right)\right)\right]$ with nonnegative initial conditions $\left[x\left(0\right),y\left(0\right),z\left(0\right)\geq 0\right]$ passing through any point ${\left(x\left(t\right),y\left(t\right),z\left(t\right)\right)}^{T}$ in $R_{+,0}^{3}$ its ω–limit set is not empty and it is a compact invariant set, i.e.,
$$\lim_{t\to \infty }\phi \left(x\left(t\right),y\left(t\right),z\left(t\right)\right)={\left(0,0,0\right)}^{T},$$
see Lemma 4.1 by Khalil in [59] at Section 4.2 and Theorem 1 by Perko in [61] at Section 3.2.

Concerning the existence and uniqueness of solutions for the KM system (Equation 1)–(Equation 3), let us introduce the following notations for the sake of simplicity
$$\begin{array}{ccc} f_{1}\left(t,x,y,z\right) & = & \frac{\rho_{1}xy}{\rho_{2}+y}-\rho_{3}xz-\rho_{4}x,\\ f_{2}\left(t,x,y,z\right) & = & -\rho_{5}xy-\rho_{6}yz-\rho_{7}y,\\ f_{3}\left(t,x,y,z\right) & = & \rho_{8}xz+\rho_{9}yz-\rho_{10}z, \end{array}$$
and compute the Jacobian matrix $\left[\partial f/\partial u\right]\left(t,u\right)$ (see [49] at Section 7.4) with results shown below for $f_{i}\left(t,u\right),i=1,2,3;$ and $u={\left[x,y,z\right]}^{T}$
(16)$$J=\left[\begin{array}{ccc} \frac{\rho_{1}y}{\rho_{2}+y}-\rho_{3}z-\rho_{4} & \frac{\rho_{1}\rho_{2}x}{{\left(\rho_{2}+y\right)}^{2}} & -\rho_{3}x\\ -\rho_{5}y & -\rho_{5}x-\rho_{6}z-\rho_{7} & -\rho_{6}y\\ \rho_{8}z & \rho_{9}z & \rho_{8}x+\rho_{9}y-\rho_{10} \end{array}\right],$$
and it is evident that $f_{i}\left(t,u\right)$ and $\left[\partial f/\partial u\right]\left(t,u\right)$ are continuous and exist on the domain $\Omega =\left[t_{0},t_{f}\right]\times K_{\Gamma }$ with $\left[t_{0},t_{f}\right]\in \left[t_{0},\infty \right]$ and $K_{\Gamma }\subset R_{+,0}^{3}$ [33]. Hence, the latter implies that $f_{i}\left(t,u\right)$ is locally Lipschitz in *u* on Ω (see Lemma 3.2 by Khalil in [59] at Section 3.1). Further, each element of (Equation 16) is bounded by Theorem 1. Thus, the following can be concluded:

**Theorem 3.** 
***Existence and uniqueness.** There is a Lipschitz constant ℓ≥0 such that $∥\left[\partial f/\partial u\right]\left(t,u\right)∥\leq \ell$ on Ω. Then, $f_{i}\left(t,u\right)$ satisfies the Lipschitz condition*

$$∥f\left(t,u_{1}\right)-f\left(t,u_{2}\right)∥\leq \ell ∥u_{1}-u_{2}∥,$$

*and there exists some δ>0 such that the KM mechanistic model (Equation 1)–(Equation 3), given as $\dot{u}=\;f_{i}\left(t,u\right)$ with $u\left(t_{0}\right)=u_{0}$, has a unique solution over $\left[t_{0},t_{0}+\delta \right]$.*


Although conditions for asymptotic stability of the equilibrium point $\left(x_{0}^{*},y_{0}^{*},z_{0}^{*}\right)=\left(0,0,0\right)$ in $R_{+,0}^{3}$ were established in Theorem 2, it is straightforward to demonstrate its local asymptotic stability by evaluating (Equation 5) in (Equation 16) as follows
$$J_{\left(0,0,0\right)}=\left[\begin{array}{ccc} -\rho_{4} & 0 & 0\\ 0 & -\rho_{7} & 0\\ 0 & 0 & -\rho_{10} \end{array}\right],$$
where the eigenvalues $\left[\lambda_{i},i=1,2,3\right]$ are given by each element of the diagonal. Thus, $\lambda_{1}=-\rho_{4}$, $\lambda_{2}=-\rho_{7}$, and $\lambda_{3}=-\rho_{10}$. Therefore, Theorem 4.7 by Khalil in [59] allows us to conclude the next additional result to Theorem 2:

**Corollary 1.** ***Local stability.** The equilibrium point $\left(x_{0}^{*},y_{0}^{*},z_{0}^{*}\right)=\left(0,0,0\right)$ of the KM mechanistic model (Equation 1)–(Equation 3) is locally asymptotically stable in* $R_{+,0}^{3}$.

## 4. Discussion

The KM mechanistic model (Equation 1)–(Equation 3) was formulated by considering the biological relationships between each variable in a controlled batch fermentation where concentrations in g/L were measured for biomass $\left[x\left(t\right)\right]$, glucose $\left[y\left(t\right)\right]$, and ethanol $\left[z\left(t\right)\right]$ over 13 consecutive hours. Then, by means of the *lsqcurvefit* function, an algorithm was developed in Matlab to approximate the experimental data from the 17 *K. marxianus* strains discussed at Section 2; both qualitative (see Figure 2, Figure 3, Figure 4, Figure 5 and Figure 6) and quantitative (see Table 3 and Table 4) results were shown in Section 3. The in silico experimentation illustrates the capabilities of the system to approximate the experimental data of each strain, whilst both the $R^{2}$ and the AIC provide a value for the goodness of fit of the model to each set of data. In Table 4, one can see that $R^{2}$ values range from 0.955 to 0.994, and AIC from −43.478 to 33.184, these values are for strains 7 and 9, respectively.

Now, it should be noted that the dynamics between biomass growth, substrate consumption and product generation have been modeled before by means of the logistic growth law [62], the Pirt Equation [63], and Luedeking–Piret Equation [64] as indicated below in Equations (Equation 17)–(Equation 19), respectively:(17)$$\begin{array}{ccc} \dot{X} & = & \mu_{\max}X\left(1-\frac{X}{X_{\max}}\right), \end{array}$$(18)$$\begin{array}{ccc} \dot{S} & = & -\frac{1}{Y_{X/S}}\dot{X}-mX, \end{array}$$(19)$$\begin{array}{ccc} \dot{P} & = & \alpha \dot{X}+\beta X, \end{array}$$
where $\mu_{\max}$ is the biomass maximum growth rate, this parameter is equivalent to $\rho_{1}$ in our mathematical model; $X_{\max}$ the maximum concentration value of biomass in the experimental data set for the time-interval of the process being observed; $Y_{X/S}$ the biomass/substrate yield; *m* is the maintenance coefficient; α is the growth-associated coefficient for the product; and β is the non-growth-associated coefficient for the product. Our algorithm was applied to approximate the experimental data of the 17 *K. marxianus* strains with overall results shown in Table 5.

The main comparison between the KM system (Equation 1)–(Equation 3) and Equations (Equation 17)–(Equation 19) is performed with respect to the biomass maximum growth rate, given by $\rho_{1}$ and $\mu_{\max}$, respectively. Table 2 and Table 5 show that estimated values of $\rho_{1}$ are on average ∼0.717 smaller than those estimated for $\mu_{\max}$. The latter is a direct consequence of the biological assumptions on which each mechanistic model was formulated. The KM system (Equation 1)–(Equation 3) was constructed by considering interactions between the three variables as illustrated in the flow diagram of Figure 1, whilst the logistic, Pirt, and Luedeking–Piret Equations (Equation 17)–(Equation 19) are constructed by only assuming a logistic growth for biomass without taking into account the overall effect of ethanol production over the entire system as well as the death rate of biomass $\left[x\left(t\right)\right]$, decomposition rate of glucose $\left[y\left(t\right)\right]$, and degradation of ethanol $\left[z\left(t\right)\right]$. Further, the in silico experimentation concerning Equations (Equation 17)–(Equation 19) illustrated in Figure A1, Figure A2, Figure A3, Figure A4 and Figure A5 at Appendix C shows that approximated values for substrate $\left[S\left(t\right)\right]$, i.e., glucose, becomes negative as time increases, which is not biologically possible for this variable. Further, one can see from the experimental data that ethanol production does not follow a smooth sigmoidal growth, the data even illustrates degradation among some strains, which is better approximated by our model as it is shown in the lower panels of Figure 2, Figure 3, Figure 4, Figure 5 and Figure 6.

When comparing the goodness of fit by computing the AIC and $R^{2}$, it is evident that the KM system (Equation 1)–(Equation 3) had overall better results than the logistic, Pirt, and Luedeking–Piret Equations (Equation 17)–(Equation 19). Although the latter has fewer parameters than ours (six and ten, respectively) and the AIC penalizes a model with more parameters to be fitted, results for the RSS were lower for the KM system which ultimately worked in our favor. Further, the capabilities of the KM mechanistic model may extend beyond its ability to approximate experimental data and estimate the biomass maximum growth rate, in Appendix D the in silico experimentation illustrates the dynamics for $t\in \left[0,39\right]$, i.e., three times the period for the experimental data. Figure A6, Figure A7, Figure A8, Figure A9 and Figure A10 show that as time increases and the substrate is no longer added into the system, then the death of biomass and degradation of ethanol begins to take over the system. The latter was expected from the asymptotic stability results of Section 3, particularly Theorem 2 and Corollary 1, as these state that the concentration of all variables will eventually be zero, i.e., both biomass $\left[x\left(t\right)\right]$ and ethanol $\left[z\left(t\right)\right]$ concentrations are going to be depleted. Additionally, it is important to note that all solutions of the KM system are bounded from above, which is consistent with the localizing domain results of Theorem 1.

Regarding the values of parameters *m* and β, our algorithm yielded results in the magnitude of $10^{-14}$ for *m* in all strains; in fact, setting *m* to zero does not affect the ultimate results for the other parameters $\left[\mu_{\max},Y_{X/S},\alpha ,\;\mathrm{and}\;\beta \right]$ which may allow us to completely disregard this term $\left[-mX\right]$ from Equations (Equation 17)–(Equation 19). Concerning β, values for 12 strains were in the same order of magnitude $\left[10^{-14}\right]$, however, the following results were determined for strains 3, 4, 11, 12, and 17: $87.633\times 10^{-3}$, $38.660\times 10^{-3}$, $51.050\times 10^{-3}$, $53.813\times 10^{-3}$, $37.702\times 10^{-3}$, respectively. Hence, the non-growth-associated coefficient for the product may influence the dynamics in some karyotypes of *K. marxianus*.

## 5. Conclusions

Mechanistic modeling has proven to be a powerful tool capable of describing the relationships between different variables in the dynamics of biological systems when considering assumptions based on scientific principles underlying the phenomenon being modeled. In this work, a set of three coupled first-order ODEs was formulated which can approximate experimental changes over time of alcoholic fermentation in batch culture by 17 different strains of *K. marxianus*.

The KM mechanistic model (Equation 1)–(Equation 3) describes biomass growth $\left[x\left(t\right)\right]$, glucose consumption $\left[y\left(t\right)\right]$, and ethanol production $\left[z\left(t\right)\right]$ in concentrations of g/L per hour. The parameter values of the system were estimated through a nonlinear curve-fitting algorithm in Matlab with the experimental data of each batch culture fermentation described in Section 2. The latter allowed us to conclude that seven parameters have the same numerical value for the dynamics observed in the 17 strains, particularly the affinity with substrate constant $\left[\rho_{2}\right]$, inhibition rate of biomass growth due to product accumulation $\left[\rho_{3}\right]$, biomass death rate $\left[\rho_{4}\right]$, consumption rates for biomass growth and ethanol production $\left[\rho_{5}\;\mathrm{and}\;\rho_{6}\right]$, glucose spontaneous decomposition rate $\left[\rho_{7}\right]$, and ethanol degradation rate $\left[\rho_{10}\right]$; these values are shown in Table 1. However, the biomass maximum growth rate $\left[\rho_{1}\right]$, ethanol production associated with biomass growth $\left[\rho_{8}\right]$, and glucose converted in ethanol $\left[\rho_{9}\right]$ parameters have specific values for each strain, results are shown in Table 2 with a 95% confidence interval that gives us the margin of error for each parameter value estimation.

As predictive microbiology establishes, mathematical models must be simplified until measurable parameters can be obtained, the KM mechanistic model successfully achieves this with $\rho_{1}$, $\rho_{8}$, and $\rho_{9}$ as the main parameters that describe the overall dynamics of the batch fermentation process under study in this research. The biomass growth rate is a very specific value for each strain that must be as high as possible. Ethanol production with respect to biomass growth represents the fermentative capacity of each strain, and the concentration of glucose converted to ethanol is directly related to these rates. It should be noted that in batch culture the latter requires high sugar concentrations to achieve alcoholic fermentation.

Further, the in silico experimentation illustrates that our model may be able to accurately predict the concentration of each variable as it is shown in Appendix D; nonetheless, further experimental data are needed to properly validate this assessment. One can see in Figure A6, Figure A7, Figure A8, Figure A9 and Figure A10 that when no more substrate is added to the culture, then biomass growth goes into the death phase, and ethanol degradation begins to happen in the system. This behavior is to be expected as the nonlinear analysis of the system allowed us to conclude that all concentrations will eventually go to zero in the absence of glucose, i.e., the asymptotic stability of the equilibrium (Equation 5) $\left[\left(x_{0}^{*},y_{0}^{*},z_{0}^{*}\right)=\left(0,0,0\right)\right]$ by Theorem 2 and Corollary 1; further, concentrations over time of all variables are bounded by the Localizing Domain Theorem 1. The latter is illustrated in all panels for the predictions of biomass growth $\left[x\left(t\right)\right]$, glucose consumption $\left[y\left(t\right)\right]$, and ethanol production $\left[z\left(t\right)\right]$.

Finally, the KM mechanistic model may be useful in the field of predictive microbiology, particularly in alcoholic fermentation through yeast and sugar, such as *K. marxianus* and glucose as only three parameters of our system needs to be fitted for different strains. Furthermore, when comparing the results of the biomass maximum growth rate of our model with the classic logistic, Pirt, and Luedeking–Piret Equations (Equation 17)–(Equation 19), our values are on average 71.7% smaller as the KM system (Equation 1)–(Equation 3) takes into account the effect of both substrate and product concentrations in the batch culture over the biomass growth phases.

## Figures and Tables

**Figure 1 entropy-25-00497-f001:**
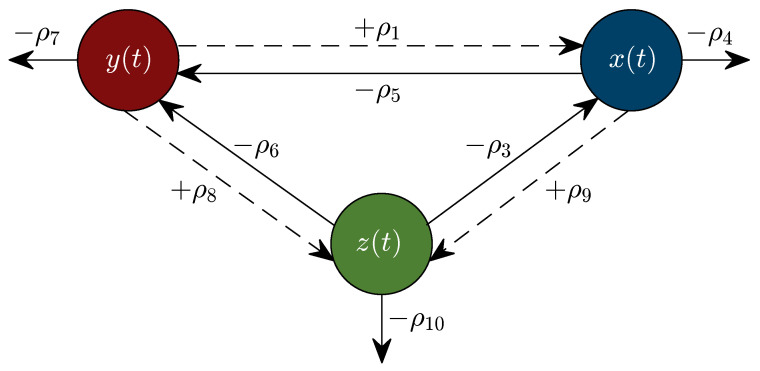
Flow diagram describing interactions between each variable and their corresponding relationship with each parameter.

**Figure 2 entropy-25-00497-f002:**
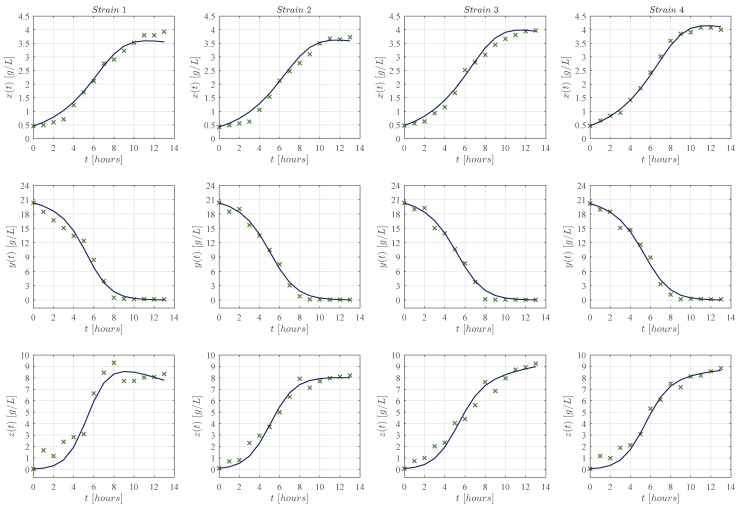
Each column from left to right (in landscape orientation) illustrates both the experimental data (× green marker), and the approximated values obtained with the KM system (continuous blue line) for each corresponding strain 1–4; the top row shows results for biomass $\left[x\left(t\right)\right]$, the middle row for glucose $\left[y\left(t\right)\right]$, and the lower row for ethanol $\left[z\left(t\right)\right]$. The × green marker represents the average value calculated from the two measurements that were made for each variable in every strain.

**Figure 3 entropy-25-00497-f003:**
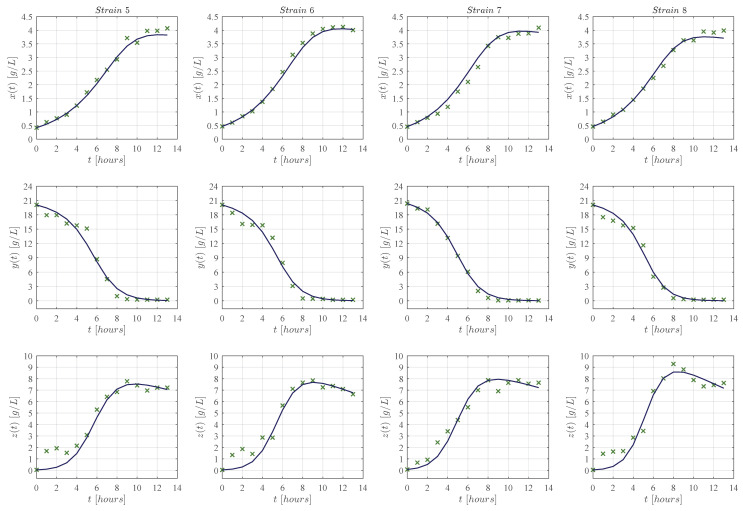
Each column from left to right (in landscape orientation) illustrates both the experimental data (× green marker), and the approximated values obtained with the KM system (continuous blue line) for each corresponding strain 5–8; the top row shows results for biomass $\left[x\left(t\right)\right]$, the middle row for glucose $\left[y\left(t\right)\right]$, and the lower row for ethanol $\left[z\left(t\right)\right]$. The × green marker represents the average value calculated from the two measurements that were made for each variable in every strain.

**Figure 4 entropy-25-00497-f004:**
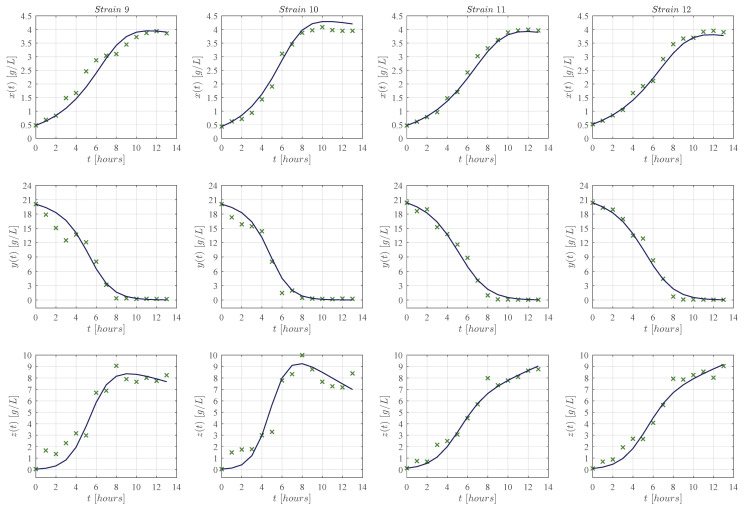
Each column from left to right (in landscape orientation) illustrates both the experimental data (× green marker), and the approximated values obtained with the KM system (continuous blue line) for each corresponding strain 9–12; the top row shows results for biomass $\left[x\left(t\right)\right]$, the middle row for glucose $\left[y\left(t\right)\right]$, and the lower row for ethanol $\left[z\left(t\right)\right]$. The × green marker represents the average value calculated from the two measurements that were made for each variable in every strain.

**Figure 5 entropy-25-00497-f005:**
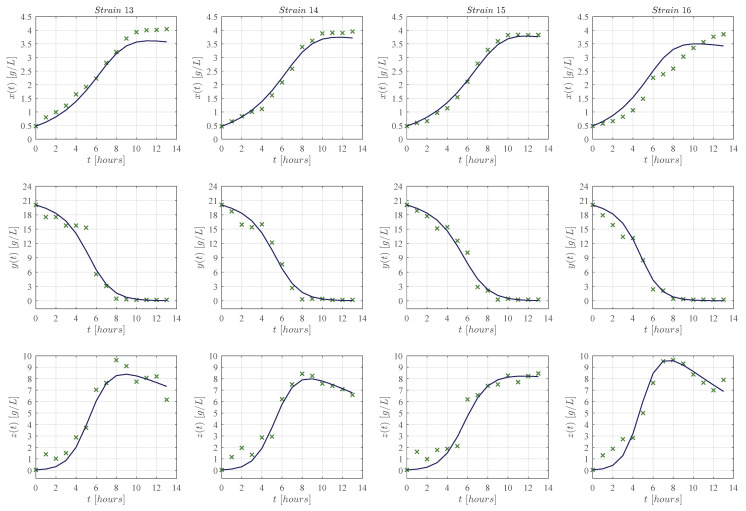
Each column from left to right (in landscape orientation) illustrates both the experimental data (× green marker), and the approximated values obtained with the KM system (continuous blue line) for each corresponding strain 13–16; the top row shows results for biomass $\left[x\left(t\right)\right]$, the middle row for glucose $\left[y\left(t\right)\right]$, and the lower row for ethanol $\left[z\left(t\right)\right]$. The × green marker represents the average value calculated from the two measurements that were made for each variable in every strain.

**Figure 6 entropy-25-00497-f006:**
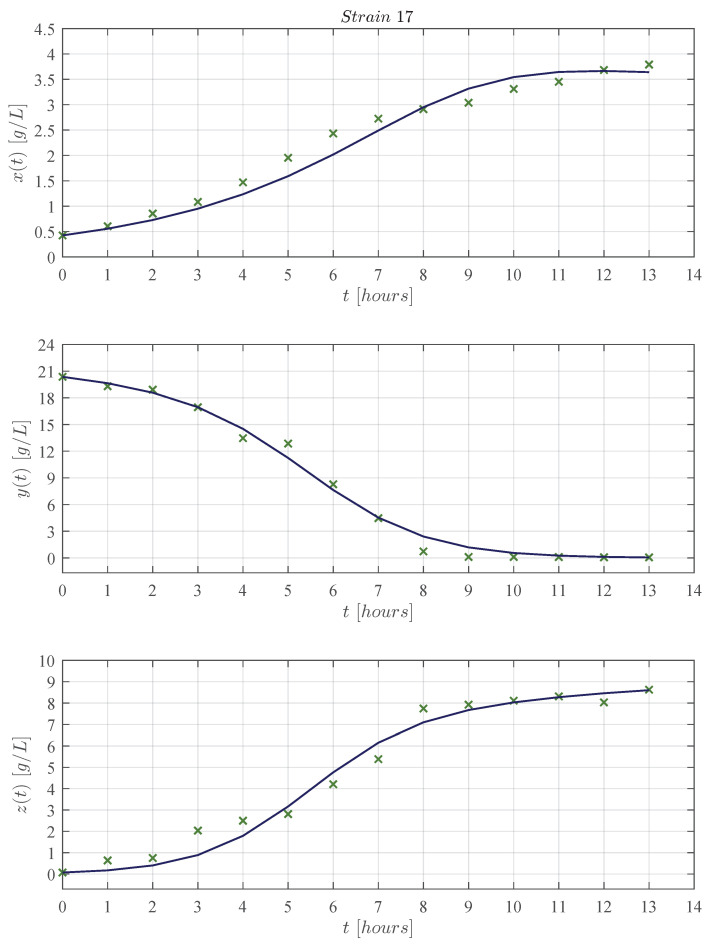
Experimental data $\left(\times \mathrm{green}\;\mathrm{marker}\right)$, and approximated values obtained with the KM system $\left(\mathrm{continuous}\;\mathrm{blue}\;\mathrm{line}\right)$ for strain 17; the top panel shows results for biomass $\left[x\left(t\right)\right]$, the panel row for glucose $\left[y\left(t\right)\right]$, and the lower panel for ethanol $\left[z\left(t\right)\right]$. The × green marker represents the average value calculated from the two measurements that were made for each variable in every strain.

**Table 1 entropy-25-00497-t001:** Description, values, and units of variables and parameters for the KM mechanistic model.

Variables/ Parameters	Description	Values	Units
$x\left(t\right)$	Biomass concentration	−	g/L
$y\left(t\right)$	Glucose concentration	−	g/L
$z\left(t\right)$	Ethanol concentration	−	g/L
$\rho_{1}$	Biomass maximum growth rate	$\left[289.385,381.419\right]\times 10^{-3}$	h${}^{-1}$
$\rho_{2}$	Affinity with substrate constant	2.281	g/L
$\rho_{3}$	Inhibition rate of biomass growth due to product accumulation	$1.066\times 10^{-3}$	L/(g × h)
$\rho_{4}$	Biomass death rate	$7.275\times 10^{-3}$	h${}^{-1}$
$\rho_{5}$	Consumption rate for biomass growth	$56.893\times 10^{-3}$	L/(g × h)
$\rho_{6}$	Consumption rate for ethanol production	$71.842\times 10^{-3}$	L/(g × h)
$\rho_{7}$	Glucose spontaneous decomposition rate	$824.233\times 10^{-9}$	h${}^{-1}$
$\rho_{8}$	Ethanol production associated with the biomass growth rate	$\left[19.088,49.816\right]\times 10^{-3}$	L/(g × h)
$\rho_{9}$	Glucose converted in ethanol	$\left[46.352,70.349\right]\times 10^{-3}$	L/(g × h)
$\rho_{10}$	Ethanol degradation rate	$149.899\times 10^{-3}$	h${}^{-1}$

**Table 2 entropy-25-00497-t002:** Fitted values, their standard error $\left[SE\left(\rho_{k}\right)\right]$, and 95% confidence intervals $\left[CI\left(\rho_{k}\right)\right]$ for the biomass growth rate $\left[\rho_{1}\right]$, and ethanol production rates $\left[\rho_{8},\rho_{9}\right]$, where all values are written with a magnitude of $10^{-3}$. Thus, it is possible to identify both lower and upper bounds for the values of the three fitted parameters as follows $\rho_{1}\in \left[289.385,381.419\right]\times 10^{-3}$, $\rho_{8}\in \left[19.088,49.816\right]\times 10^{-3}$, and $\rho_{9}\in \left[46.352,70.349\right]\times 10^{-3}$.

Strain	$\rho_{1}$	$\mathrm{SE}\;\left(\rho_{1}\right)$	95% $\mathrm{CI}\;\left(\rho_{1}\right)$	$\rho_{8}$	$\mathrm{SE}\;\left(\rho_{8}\right)$	95% $\mathrm{CI}\;\left(\rho_{8}\right)$	$\rho_{9}$	$\mathrm{SE}\;\left(\rho_{9}\right)$	95% $\mathrm{CI}\;\left(\rho_{9}\right)$
1	312.378	14.457	$\left(283.137,341.620\right)$	31.563	3.669	$\left(24.140,38.985\right)$	60.706	1.151	$\left(58.378,63.033\right)$
2	320.848	8.407	$\left(303.842,337.853\right)$	40.073	2.395	$\left(35.228,44.919\right)$	51.253	0.650	$\left(49.937,52.569\right)$
3	315.502	9.267	$\left(296.756,334.247\right)$	42.560	2.657	$\left(37.186,47.935\right)$	52.146	0.797	$\left(50.533,53.759\right)$
4	319.364	8.448	$\left(302.276,336.453\right)$	38.795	2.368	$\left(34.005,43.585\right)$	54.576	0.763	$\left(53.032,56.119\right)$
5	312.618	13.016	$\left(286.291,338.945\right)$	30.025	3.939	$\left(22.056,37.993\right)$	57.305	1.125	$\left(55.030,59.580\right)$
6	318.556	12.204	$\left(293.870,343.242\right)$	25.252	3.360	$\left(18.456,32.048\right)$	60.595	1.126	$\left(58.316,62.874\right)$
7	336.119	7.376	$\left(321.200,351.038\right)$	29.110	1.793	$\left(25.483,32.737\right)$	56.806	0.627	$\left(55.538,58.075\right)$
8	326.840	11.939	$\left(302.691,350.989\right)$	25.102	2.794	$\left(19.450,30.754\right)$	64.465	1.004	$\left(62.434,66.497\right)$
9	322.375	17.757	$\left(286.457,358.293\right)$	29.314	4.383	$\left(20.448,38.180\right)$	61.948	1.570	$\left(58.771,65.125\right)$
10	381.419	15.931	$\left(349.195,413.642\right)$	19.088	3.051	$\left(12.915,25.260\right)$	69.068	1.414	$\left(66.208,71.928\right)$
11	307.816	9.485	$\left(288.631,327.002\right)$	48.803	3.070	$\left(42.593,55.014\right)$	46.352	0.810	$\left(44.714,47.990\right)$
12	289.385	10.222	$\left(268.709,310.060\right)$	49.816	3.380	$\left(42.978,56.654\right)$	48.047	0.887	$\left(46.252,49.841\right)$
13	309.540	17.851	$\left(273.434,345.647\right)$	28.475	4.498	$\left(19.376,37.574\right)$	62.432	1.481	$\left(59.436,65.428\right)$
14	312.244	12.998	$\left(285.953,338.536\right)$	25.011	3.442	$\left(18.050,31.973\right)$	61.883	1.121	$\left(59.615,64.151\right)$
15	298.551	11.822	$\left(274.638,322.464\right)$	37.245	3.440	$\left(30.287,44.202\right)$	57.670	1.036	$\left(55.574,59.766\right)$
16	335.425	16.486	$\left(302.078,368.773\right)$	20.480	3.266	$\left(13.873,27.088\right)$	70.349	1.295	$\left(67.728,72.969\right)$
17	310.122	9.556	$\left(290.793,329.451\right)$	44.563	2.980	$\left(38.534,50.592\right)$	50.780	0.755	$\left(49.252,52.307\right)$

**Table 3 entropy-25-00497-t003:** The $R^{2}$ provides a measure of how well the experimental data are replicated by the KM mathematical model (Equation 1)–(Equation 3) for each strain. This coefficient was computed independently for each variable, i.e., biomass $\left[x\left(t\right)\right]$, glucose $\left[y\left(t\right)\right]$, and ethanol $\left[z\left(t\right)\right]$. One can see that the values for $R^{2}$ ranges between 0.902 to 0.997 which allows us to conclude an overall well goodness of fit for the model.

Strain	Biomass	Glucose	Ethanol
1	0.980	0.979	0.923
2	0.983	0.994	0.974
3	0.983	0.990	0.967
4	0.996	0.990	0.974
5	0.989	0.979	0.926
6	0.995	0.979	0.940
7	0.982	0.997	0.960
8	0.990	0.985	0.951
9	0.953	0.950	0.918
10	0.979	0.972	0.905
11	0.992	0.988	0.974
12	0.985	0.988	0.967
13	0.960	0.962	0.942
14	0.986	0.979	0.947
15	0.992	0.983	0.945
16	0.902	0.974	0.940
17	0.965	0.991	0.973

**Table 4 entropy-25-00497-t004:** In order to provide overall measures for the fitting capabilities of our mathematical model, i.e., the KM system (Equation 1)–(Equation 3), values were calculated for the RSS, the AIC, and the $R^{2}$ to estimate and describe the dynamics between the three variables $\phi \left(x,y,z\right)$, where $x\left(t\right)$, $y\left(t\right)$ and $z\left(t\right)$ represent, respectively, the evolution of biomass, glucose, and ethanol.

Strain	RSS	AIC	$R^{2}$
1	28.375	+10.626	0.976
2	9.233	−36.530	0.993
3	13.679	−20.020	0.989
4	12.267	−24.596	0.990
5	26.471	+7.709	0.979
6	24.403	+4.292	0.980
7	7.825	−43.478	0.994
8	19.638	−4.832	0.983
9	48.551	+33.184	0.954
10	37.663	+22.520	0.966
11	13.978	−19.111	0.989
12	16.201	−12.912	0.988
13	43.020	+28.105	0.966
14	24.183	+3.912	0.980
15	21.902	−0.248	0.983
16	30.653	+13.869	0.972
17	13.047	−22.005	0.990

**Table 5 entropy-25-00497-t005:** The logistic, Pir, and Luedeking–Piret Equations (Equation 17)–(Equation 19) provides valuable information concerning biomass growth $\left[\mu_{\max}\right]$, biomass/substrate yield $\left[Y_{X/S}\right]$, and product generation $\left[\alpha \right]$; estimated numerical values are given in their respective columns. Concerning the goodness of fit, results regarding the RSS, AIC, and $R^{2}$ are provided in the following columns.

Strain	$\mu_{\max}\left[\times 10^{-3}\right]$	$Y_{X/S}\left[\times 10^{-3}\right]$	α	RSS	AIC	$R^{2}$
1	442.669	159.066	2.648	35.905	+7.815	0.970
2	451.284	150.295	2.551	19.641	−17.523	0.984
3	437.516	158.265	1.903	23.501	−9.987	0.982
4	428.145	164.594	2.185	25.859	−5.971	0.979
5	417.879	166.067	2.296	50.863	+22.441	0.960
6	434.735	169.347	2.253	48.715	+20.629	0.959
7	482.158	168.159	2.248	21.992	−12.774	0.982
8	463.808	165.934	2.505	47.813	+19.844	0.959
9	455.745	163.157	2.530	40.974	+13.362	0.961
10	544.441	175.477	2.344	55.484	+26.094	0.950
11	422.190	158.260	2.120	23.199	−10.530	0.981
12	394.033	151.888	2.241	34.084	+5.629	0.974
13	433.945	165.334	2.562	83.294	+43.158	0.935
14	437.632	161.878	2.422	52.732	+23.957	0.956
15	394.825	151.920	2.692	42.685	+15.079	0.966
16	523.096	162.308	2.643	41.717	+14.116	0.962
17	421.708	150.435	2.352	28.825	−1.411	0.978

## Data Availability

The data supporting the findings of the present work are available through authors Nicolás O. Soto-Cruz and Jesús B. Páez-Lerma upon reasonable request.

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
