# Peer review of "Mechanistic Modelling of Biomass Growth, Glucose Consumption and Ethanol Production by Kluyveromyces marxianus in Batch Fermentation"

_entropy, 2023, doi:10.3390/e25030497_

Round 1

Reviewer 1 Report

The article presents an interesting and valuable study related to modeling of  batch culture fermentation by Kluyveromyces marxianus. A very good impression is made by the combination of theoretical (a complete analysis of the local and global dynamics of proposed biological system by nonlinear theory), identification (optimization) and simulation  (prediction) studies, which gives  completeness to the article. Modeling results are evaluated both qualitatively by in silico experimentation and quantitatively by the coefficient of determination and Akaike's information criterion in the basis of 17 sets of experimental data. The comparison of the proposed model with the classical model given by logistics, Peart and Ludeking-Piret equations is representative. It would be interesting in the future to make a comparison with other models available in the literature describing different types of inhibition by substrate and product, as well as to what extent the proposed model is valid for alcoholic fermentations with other yeast (for example Saccharomyces cerevisiae.

It is not clear how the averaged data for the groups of 4 experimental data sets were obtained. The text explains that with the × green marker are presented  the averages of two experimental sets data measurements for each variable. An explanation is needed, as well as additional information for optimization method.

Another study related to verifying the predictive properties of the proposed model would be interesting. Model identification (i.e. model parameter values) to be obtained based on several experiments. The verification should be implemented based on an experiment that is not included in the model parameter setting. In this way, it can be established whether the model can predict the dynamics of subsequent experiments.

Reviewer 2 Report

Introduction:

1. You have mentioned that batch fermentation is preferred for industrial applications because it is easy to be controlled, but there are other advantages that make this mode of fermentation more common, you can refer to the following paper to mention these advantages (https://pubs.acs.org/doi/10.1021/acsomega.0c04025).

2. It is generally hard to model batch fermentation (generally biological systems) due to the time-varying characteristics of biological systems, which often result in process nonlinearities. The multiplicity of reactions, the adaptability and evolution of organisms over short periods of time, and the continuous shift in environmental conditions are features that characterize batch processes.   It is good to highlight that in the introduction as a motivation for your research.

3. " Mathematical models in predictive microbiology are classified into primary and secondary ..."; I prefer to classify them into structured and unstructured models, pls find more details in the following paper (https://doi.org/10.3390/biom9080308).

4. Please state the novelty of your work in the paper.

5. Please delete the last section of the introduction Line 79- line 102, it seems like an outline of the thesis which I do not think should be stated in this research article.

6. Overall, try to optimize the introduction and cohesion between the sections. 

Methodology:

Table 2: Please delete " over time in hours" from the description of X(t), y(t), and z(t). 

Results

- Please set the direction of the graphs and their axis titles to be easily readable. 

- I think it will be better to include the theoretical background (section 3.2) in the appendix (optional)

Discussion:

- Please take this recommendation into account only f you considered deleting section 3.2;  I suggest combining the results and discussion sections (optional).

- Please explain the observed increase in the product concentration (ethanol)  after 10 hours of fermentation (Experimental data in Fig A1,A2, A3,A4) whereas it was predicted to be constant after the depletion of the substrate (glucose). 

- Use passive voice rather than active voice " we.................., we...............".

Conclusion:

Overall, it is fine

Reviewer 3 Report

In this contribution, the authors propose an ODE model for the growth of K. marxianus with 3 variables (biomass, glucose and ethanol) and parametrized it for 17 strains of this yeast species. In great depth, the authors have also provided a rigorous mathematical treatment for the existence and stability of feasible equilibrium points, and provided measures for the goodness of fit while comparing the model with alternative mathematical descriptions.

Unfortunately, I find this work too preliminary, with some serious issues that must be addressed by the authors before a novel submission.

The major problems are:

- The authors have completely overlooked the need to provide a sound motivation for their work and do not present a clear application for their findings and parametrization. Are these 17 strains well known and important in agricultural/industrial/biotechnological contexts? Where do they come from, and what are the differences between them that would explain the results of table 3? In summary, why is table 3 useful? The authors spend a few paragraphs of the introduction indicating the contents of the manuscript in a place where purpose and motivation are traditionally found in a scientific paper.

- The experimental section needs a lot more detail in what concerns the assays of glucose and ethanol in the growth media. HPLC was the technique, sure, but what are the analytical standards, what are the retention times and how was calibration carried out. Looking at the data, it seems that analytical variability is not too low and, for the purposes of model calibration, the quality of the data and reliability of the assays must be very high. In contrast, it is quite useless to include table 1, which carries only the information that the mean of two assays was computed for the 3 variables for each time point. We do not need to put that in a table. Finally, the nature of the optimization algorithms implemented in function lsqcurvefit should have been stated. The authors treated this function as a black-box.

- The modelling has also some issues. The terms corresponding to rho 6 and rho 9 seem to represent an interaction between glucose and ethanol and producing ethanol. That does not make sense. The term corresponding to rho 8 should only depend on biomass and not be (positively) affected by ethanol. Finally, if the half lives of glucose or ethanol decompositions are so high, then these terms should be just removed from the model. Maybe these criticisms result from the lack of proper logical support from the authors when they are describing the model.

- The predictive ability of the model is just displayed by running the simulations past the last experimental time point. This does not support the claim the model is making good predictions, as there are not any independent data or information to test the predictions, besides that fact that the asymptotic trends are confirmed.

- Overall, the structure, language and exposition of the manuscript should be redone: besides fixing the problem in line 96, the lengthy description of the manuscript contents in the introduction should be removed. At times, the authors refer to rates of change or growth in variables interchangeably with the variables themselves, and that can become quite confusing.

As a good point, the mathematical, dynamical system's analysis of the equilibria in the model is quite sound and strong, but this submission is, overall, in a very preliminary state.

Round 2

Reviewer 1 Report

Thanks to the authors for the answers. I believe that after the additions made, the article can be published.

Author Response

We greatly appreciate all comments on our revised manuscript and the time invested in our work.

Reviewer 2 Report

NA

Author Response

(The authors gave the same response as above.)

Reviewer 3 Report

It is clear that the authors have provided a revised version that addresses many of the major problems that could be found in the previous submission. The structure of the paper was improved and, quite significantly, some lengthy parts of the text that could be considered almost pointless have now been removed. Instead, the authors have now included a thorough description of different types of models and provide a lot more methodological detail, especially in what concerns chemical analysis and model fitting and model mathematical structure.

I must insist in the following two points. I believe that the manuscript can further improve if the authors properly address them:

1. The details on the genetic and phenotypical biological background of the strains is still too vague. It seems that disclosure of these details is either relegated to other papers or is contained in unpublished work. As it is so fundamental to the understanding of the choice of the authors to model the growth of these strains, even considering that this species is an alternative to S. cerevisiae, these details would clarify the scenarios where this work could actually be applied and should be brought to the forefront of the exposition. I understand that the authors first want to build a robust predictive tool that can be used in the control/optimization of growth conditions, but a clear prediction that points to the advantages of using this species and these strains should be provided, to make the motivation more clear.

2. While I (now) totally understand why the terms for the non-biological decompositions of glucose and ethanol must be present, the form of the terms corresponding to rho6 and rho9 still puzzles me. In their response, the authors present a technical justification, based on parameter significance. Parameter significance is not a substitute for model interpretability, and the great historical success of ODE models comes from their interpretability. To clarify my problem, let's focus on the conversion from glucose to ethanol. This is modelled by - rho6 x glucose x ethanol. What is the interpretation for such a term? The rate of conversion increases with glucose content and with ethanol content. This is typical of a second order transformation, where a molecular encounter between glucose and ethanol would be productive, and the result would be... another ethanol molecule. This does not make much sense to me, and can not find a good interpretation for this term. Biomass is responsible for the conversion from glucose to ethanol, and this term would make much more sense if it was - rho6 x glucose x biomass, or -rho6 x y instead of -rho6 y z. The same problem extends to the term corresponding to rho6. Can the authors provide a good interpretation for the mathematical form of these two terms?

Author Response

Please see attached PDF file.
